# Spatial EGFR Dynamics and Metastatic Phenotypes Modulated by Upregulated EphB2 and Src Pathways in Advanced Prostate Cancer

**DOI:** 10.3390/cancers11121910

**Published:** 2019-12-01

**Authors:** Yen-Liang Liu, Aaron M. Horning, Brandon Lieberman, Mirae Kim, Che-Kuang Lin, Chia-Nung Hung, Chih-Wei Chou, Chiou-Miin Wang, Chun-Lin Lin, Nameer B. Kirma, Michael A. Liss, Rohan Vasisht, Evan P. Perillo, Katherine Blocher, Hannah Horng, Josephine A. Taverna, Jianhua Ruan, Thomas E. Yankeelov, Andrew K. Dunn, Tim H.-M. Huang, Hsin-Chih Yeh, Chun-Liang Chen

**Affiliations:** 1Graduate Institute of Biomedical Sciences, China Medical University, Taichung 404, Taiwan; allen.liu@mail.cmu.edu.tw; 2Department of Biomedical Engineering, University of Texas at Austin, 107 W. Dean Keeton, BME Building, Austin, TX 78712, USA; sunnymiraekim@utexas.edu (M.K.); rohanvasisht@utexas.edu (R.V.); eperillo@utexas.edu (E.P.P.); katieblocher@utexas.edu (K.B.); thomas.yankeelov@utexas.edu (T.E.Y.); adunn@utexas.edu (A.K.D.); 3Department of Molecular Medicine, Mays Cancer Center, University of Texas Health Science Center, 8210 Floyd Curl Drive, Mail code: 8257, San Antonio, TX 78229, USA; ahorning@stanford.edu (A.M.H.); hsnu97712@gmail.com (C.-K.L.); hungc@uthscsa.edu (C.-N.H.); chouc@uthscsa.edu (C.-W.C.); wangc10@uthscsa.edu (C.-M.W.); chunlin.lin2014@gmail.com (C.-L.L.); kirma@uthscsa.edu (N.B.K.); huangt3@uthscsa.edu (T.H.-M.H.); 4Department of Biology, Trinity University, San Antonio, TX 78212, USA; liebermanb@uthscsa.edu; 5Department of Urology, University of Texas Health Science Center, San Antonio, TX 78229, USA; liss@uthscsa.edu; 6Department of Bioengineering, the University of Maryland, College Park, MD 20742, USA; hhorng07@terpmail.umd.edu; 7Department of Medicine, Mays Cancer Center, University of Texas Health Science Center, San Antonio, TX 78229, USA; tavernaj@uthscsa.edu; 8Department of Computer Science, University of Texas at San Antonio, San Antonio, TX 78249, USA; jianhua.ruan@utsa.edu; 9Institute for Computational Engineering and Sciences, University of Texas at Austin, Austin, TX 78712, USA; 10Department of Diagnostic Medicine, Dell Medical School, University of Texas at Austin, Austin, TX 78712, USA; 11Department of Oncology, Dell Medical School, University of Texas at Austin, Austin, TX 78712, USA; 12Livestrong Cancer Institutes, University of Texas at Austin, Austin, TX 78712, USA; 13Texas Materials Institute, University of Texas at Austin, Austin, TX 78712, USA

**Keywords:** prostate cancer, EGFR, EphB2, Src, epithelial–mesenchymal transition, metastasis, single-particle tracking, diffusivity, compartmentalization, actin, endocytosis

## Abstract

Advanced prostate cancer is a very heterogeneous disease reflecting in diverse regulations of oncogenic signaling pathways. Aberrant spatial dynamics of epidermal growth factor receptor (EGFR) promote their dimerization and clustering, leading to constitutive activation in oncogenesis. The EphB2 and Src signaling pathways are associated with the reorganization of the cytoskeleton leading to malignancy, but their roles in regulating EGFR dynamics and activation are scarcely reported. Using single-particle tracking techniques, we found that highly phosphorylated EGFR in the advanced prostate cancer cell line, PC3, was associated with higher EGFR diffusivity, as compared with LNCaP and less aggressive DU145. The increased EGFR activation and biophysical dynamics were consistent with high proliferation, migration, and invasion. After performing single-cell RNA-seq on prostate cancer cell lines and circulating tumor cells from patients, we identified that upregulated gene expression in the EphB2 and Src pathways are associated with advanced malignancy. After dasatinib treatment or siRNA knockdowns of EphB2 or Src, the PC3 cells exhibited significantly lower EGFR dynamics, cell motility, and invasion. Partial inhibitory effects were also found in DU145 cells. The upregulation of parts of the EphB2 and Src pathways also predicts poor prognosis in the prostate cancer patient cohort of The Cancer Genome Atlas. Our results provide evidence that overexpression of the EphB2 and Src signaling pathways regulate EGFR dynamics and cellular aggressiveness in some advanced prostate cancer cells.

## 1. Introduction

Epidermal growth factor receptor (EGFR) plays an important role in cell proliferation, survival, migration, and differentiation during development [1,2]. Dysregulated EGFR signaling is commonly found in cancers and conveys advanced aggressive malignancies: drug resistance, migration, invasion, and metastasis with poor prognosis [3]. The leading causes of the aberrant upregulated EGFR signaling have been identified as gene amplification [4], constitutive activation [5], dysregulated lateral organization and mobility of plasma membrane components [6], and abnormal intracellular trafficking [7]. The former two genetic mechanisms are well studied, but the latter two for EGFR spatial localization are only emerging as oncogenic contributors and are tightly modulated by the cytoskeleton under the tight control of cell adhesion, contact, and polarity. Additionally, since clathrin- and caveolin-mediated endocytosis controls the degradation and recycling of EGFRs [8], the deregulated EGFR endocytosis results in prolonged activation, which can also lead to the development of cancer [9,10]. Despite accumulating evidence suggesting that many types of cancers are caused by the dysregulation of spatial distribution and trafficking of EGFRs [6,11], few studies have reported real-time intracellular trafficking and dynamic signaling processes of EGFRs in live cells using single-particle/molecule tracking (SPT/SMT) techniques [12,13]. 

SPT/SMT techniques have been extremely valuable in deciphering cell membrane biophysics including receptor tyrosine kinase (RTK) dynamics [14,15]. Kusumi proposed the membrane skeleton mesh model to explain why membrane-bound EGFRs exhibit hop diffusion instead of Brownian diffusion [16]. This model further evolved into the compartmentalized fluid model in which the plasma membrane is compartmentalized into microdomains [15]. Some groups have also demonstrated that transmembrane proteins are constrained by the cortical actin network, and the dissociation of cortical actin has been correlated with the increased diffusivity of transmembrane proteins, such as EGFR [16,17,18,19], G-protein coupled receptor [20], and B cell receptor [21,22]. Moreover, spatiotemporal confinement which facilitates the oligomerization of membrane receptors and their associated molecules [17,18,23] is important for enhancing signaling in compartments [20,24,25,26]. In our previous study, we developed a phenotyping assay named Transmembrane Receptor Dynamics (TReD) and demonstrated that EGFR dynamics could serve as a biophysical marker to differentiate highly invasive breast cancer cell lines from benign and less invasive breast cell lines [27]. We found that highly metastatic cells exhibit fewer cortical actin filaments on the apical side of cells but more stress fibers on the basal side for migration, which leads to an increase of EGFR diffusivity on the apical side of the plasma membrane. However, the details of the biomolecular mechanism are still unclear.

Several proteins have been shown to regulate EGFR dynamics in cancer. Amplified MET collaborates with ErbB3 to influence the latter’s distribution, inducing ErbB3-dependent PI3K signaling and driving lung cancer resistance to EGFR inhibitors [28,29]. αE-catenin acts together with NF2 to secure adherens junctions and mediates inhibition of physical EGFR association, internalization, and signaling [30,31]. Loss of function in these two proteins leads to persistent EGFR activation in human cancer. Upregulated Calveolin-1 induced by hypoxia enhances EGFR clustering within the caveolae and stimulates ligand-independent activation [32]. Defects in E-Cadherin have displayed epithelial–mesenchymal transition (EMT) hallmarks and invasive phenotypes through RhoA GTPase in cancer and dysregulated EGFR signaling [33,34]. Moreover, the distribution of Eph receptors on the cell membranes of cancer cells and surrounding stromal cells controls tumor compartmentalization and invasion behaviors in colon and prostate cancers [35,36]. It is still not clear whether these proteins are part of a coordinating network or act independently. However, those proteins are vital components of the EphB2 and Src signaling pathways and may regulate EGFR dynamics. While EGFR, Eph receptors, and Src regulate cell motility and invasion through the Rho, Ras, and MEK pathways in actin and cytoskeleton assembly [37,38,39,40], studies on Eph-ephrin and Src signaling functions in EGFR dynamics are scarce.

In this study, we explored the connection between prostate cancer metastasis, EGFR dynamics, and underlying molecular mechanisms (Figure 1A). The metastatic potentials of prostate cancer cells were quantified by cell migration, invasion, and proliferation assays. The correlation between metastatic potential and EGFR dynamics was tested. Indeed, the highly metastatic prostate cancer cell line, PC3, exhibits significantly higher EGFR diffusivity and larger microdomains than a non-invasive (LNCaP) and less invasive (DU145) cancer cell line. From single-cell RNA-seq data and prohibition experiments, we found that the EphB2 and Src signaling pathways are strongly upregulated in PC3, which implicates their roles in the regulation of actin organization, EGFR dynamics, and metastasis.

## 2. Results

### 2.1. Increased EGFR Diffusivity is Correlated with Advanced Malignancy

To elucidate the connection between EGFR dynamics and malignancy, we firstly performed the TReD assay on EGFRs in five prostate epithelial cell lines (Figure 1B). Trajectories of 455–2961 single EGFR complexes (EGFRs recognized by anti-EGFR IgG antibody-conjugated fluorescent nanoparticles) from each cell line were analyzed using a modified mean-squared displacement (MSD) algorithm [41,42] to calculate EGFR diffusivity (*D*) and size of the linear confinement (*L*) [12] (Figure 1B and Appendix A, Method S1 and S2). From our tracking results, PC3 cells had the highest *D* and the largest *L*. In particular, the EGFR diffusivity of PC3 cells (0.0177 ± 0.0011 µm^2^/s, n = 1298, the statistic estimators represent sample mean ± standard error of the mean) was 144% (*p* < 0.00001), 124% (*p* < 0.00001), 97% (*p* < 0.00001), higher than those of BPH1 (0.0073 ± 0.0006 µm^2^/s, n = 928), LNCaP (0.0080 ± 0.0008 µm^2^/s, n = 455), and DU145 (0.0090 ± 0.0004 µm^2^/s, n = 2961), respectively (Figure 1B). As distinguishable as diffusivity, *L* of PC3 cells (132.1 ± 4.5 nm, n = 1298) was 59% (*p* < 0.00001), 43% (*p* < 0.00001), 35% (*p* < 0.00001) larger than those of BPH1 (83.1 ± 4.0 nm, n = 928), LNCaP (92.4 ± 6.3 nm, n = 455), and DU145 (97.8 ± 2.1 nm, n = 2961) (Appendix A), respectively. Around 15–20 trajectories were collected from every single cell, and at least 30 cells were tested for each cell line. The clear differentiation of the metastatic PC3 cells from the LNCaP and DU145 was remarkable evidence that the changes of EGFR dynamics appear to be associated with metastatic characteristics. Interestingly, LNCaP-Abl exhibits a slight increase in EGFR dynamics, which indicates the androgen-deprivation might select an advanced subpopulation from the parental LNCaP (Figure 1B) [43].

Since motility and invasion are the hallmarks of advanced cancer cells, we conducted assays to quantify migration, invasion, and proliferation of LNCaP, DU145, and PC3 using an IncuCyte system (Figure 1C–F). Consistent with the previous studies, LNCaP appeared less malignant in three assays. Distinct kinetic trends of the two advanced metastatic and castration resistant prostate cancer lines, PC3 and DU145, appeared around 48 h (46.0 relative wound density (RWD)% and 35.3 RWD% at the 48th hour, *p* = 0.0059) [43] and reached the maximal difference at 84–96 h (Figure 1C), although DU145 proliferation rate is about the same as LNCaP (Figure 1D). Among the three cell lines, these three phenotypes were positively correlated with their EGFR dynamics (*D*, mm^2^/s) and the Pearson correlation coefficients between averaged EGFR diffusivity and proliferation, migration, and invasion are 0.96, 0.92, and 0.82, respectively (Figure 1D–F). Moreover, we further explored the potential of EGFR internalization to characterize the advanced prostate cancer cells and developed a 3D inward movement assay that combines 3D single-particle tracking and 2P scanning imaging to monitor the internalization of fluorescently labeled EGFR in living cells [44] (Appendix A, Method S3–S6). We quantified the EGF-stimulated EGFR internalization by measuring the displacement of inward movement and the velocity of EGFR internalization (Appendix A). Compared to LNCaP, the highly metastatic PC3 cells exhibited not only longer time-weighted inward movement (2.83 ± 0.05 µm vs. 1.45 ± 0.04 µm, *p* < 0.0001) and higher velocity (0.021 ± 0.004 µm/s vs. 0.005 ± 0.001 µm/s, *p* = 0.0008). Like diffusivity and compartmentalization, EGFR internalization dynamics of LNCaP-Abl is in a moderate status between LNCaP and the more advanced PC3. These results imply the active EGFR internalization might also be correlated with advanced malignancy. As several studies recently revealed mechanical regulations of receptor tyrosine kinase (RTK), including EGFR signaling in cancers [6,7,45,46], these together imply roles of EGFR dynamics in advanced malignancy in prostate cancer and their potential as biophysical markers for this disease.

### 2.2. Depolymerization of Cortical Actin Matrix Increased EGFR Diffusivity and Enlarged Membrane Compartments

Previous studies have shown that transmembrane protein dynamics may be regulated by actin organization [6,7,47]. To investigate how the cortical actin structure modulates EGFR dynamics, we treated the three cell lines with Latrunculin B (LatB) which depolymerizes actin network [48]. The images of actin filaments were acquired by a super-resolution microscope, Structured Illumination Microscopy (SIM). The EGFR dynamics on the apical side of the plasma membrane of cells was measured by the 2D-SPT assay. Intact LNCaP and DU145 exhibited a denser actin meshwork on the apical membranes and abundant peri-junctional actin bands (yellow arrowheads in Figure 2A). In contrast, the two more invasive cell lines exhibited more parallel actin filaments at the basal side of cells (yellow arrows in Figure 2A). The xz orthogonal projections clearly showed that LNCaP displays the densest content of actin at the apical side of three cell lines (Figure 2B,C). Upon LatB treatment, LNCaP cells lost peri-junctional actin bands and cortical actin networks along their apical plasma membrane (Figure 2B). In addition, the LatB-treated DU145 and PC3 cells exhibited more dot-like actin rather than fibrous actin (zoomed-in insets in Figure 2A). These results indicated that the LatB substantially disrupted stress fibers, reduced cortical actin, and retracted filopodia (red arrowheads in Figure 2A), and caused a decrease in projected cell area observed in all three cell lines (Figure 2A). The xz projections clearly showed the dissociation of cortical actin from the apical surface of the plasma membrane after LatB treatment (Figure 2B,C). The detailed information of the quantification of cortical actin is shown in Appendix A. This disassembled cortical actin meshwork may be responsible for the increased diffusivity of the EGFR complex in the LatB-treated LNCaP and DU145 cells (Figure 2D) and the enlarged confinement size (Figure 2E). Interestingly, although LatB depolymerized the F-actin in PC3 cells, mainly, the stress fibers at the basal side of the cells (Figure 2A), PC3’s EGFR diffusivity decreased, and the confinement size remained in the same level (Figure 2D,E). We speculated that the decrease of EGFR diffusivity is due to the reduction of the surface area of the plasma membrane of PC3. The actin network perturbation data showed that metastatic potentials of cell lines are inversely correlated with densities of cortical actin beneath the apical membranes. Denser cortical actin in LNCaP and DU145 reduced EGFR diffusivities. Therefore, the TReD assay might hold a promise to quantify metastatic potential by measuring EGFR dynamics.

### 2.3. Upregulated EphB2 and Src Pathways In Cell Lines, CTCs, and Tumors Predicted Actin Reorganization, Metastasis, and Poor Prognosis

As advanced prostate cancer cells exhibit higher EGFR diffusivity, larger compartments of the plasma membrane, and loose cortical actin networks, we were interested in investigating the underlying molecular mechanisms for the differential actin organization. From single-cell RNA-seq data of LNCaP, PC3, and LNCaP-Abl (Horning et al., unpublished data), we found 2395 differentially expressed (DE) genes and 146 enriched pathways correlated with advanced prostate cancer. The EphB2 and Src pathways are of particular interest to us due to their involvement in the regulation of RAS signaling, cytoskeletal organization, actin assembly, EMT, metastasis, and differential expression (Figure 3A,B and Appendix A) [45,49,50,51]. The Eph-ephrin (n = 20) and Src (n = 171) pathways showed an extensive overlapping of DE genes (65%, *p* = 1.247 × 10^−27^, labeled with asterisks in Appendix A), indicating high convergence in their functions. The EphB2 forward signaling pathway (n = 12) was identified as well, suggesting that it has a significant contribution in the Eph-ephrin pathway (25%, *p* = 1.56 × 10^−8^, double asterisks) (Appendix A). To probe the robust nature of scRNA-seq data, the expression of representative DE genes (n = 16) were validated in four cell lines using specific primers (Appendix A) and qRT-PCR [43]. Overall results were consistent with single-cell RNA-seq data, whereas LNCaP-Abl and DU145 were moderate in some sets of genes indicating less advanced malignancy. 

The majority of DE genes in Src, Eph-ephrin, and EphB-mediated pathways (85%, 75%, and 83%) were upregulated in PC3, as compared to the other two cell lines that shared similar expression patterns (Figure 3A,B and Appendix A). Based on the gene expression patterns of EphB2 forward and EPH-Ephrin pathways, LNCaP and LNCaP-Abl were grouped using unsupervised hierarchical clustering. PC3’s gene expression pattern implicated an activated mode for EMT, F-actin reprogramming, endocytosis, cell motility, and invasion. Those highly expressed genes are the major players in metastatic pathways according to DAVID gene ontology analysis and Genecards database mining [52,53]. RTKs (*EPHB2*, *EPHA2, TGFR2*, and *AXL*) and their ligands (*EFNB1* and *EFNB3*) lead to activation of downstream cytoplasmic kinases (*FYN*, *PTK2 (FAK)*, *PKC2*, *PTK2B*, *PAK6*, *MAPK3*, and *MAP4K4*) and the Ras signaling pathway (*RAC1*, *ROCK2*, *ARHGEF28*, *RAP1B*, *ARHGDIA*, *ARHGDIB*, and *ARAF*) (Appendix A). Many upregulated genes are involved in actin organization (*ARPC2*, *ARPC1A*, *AFAP1*, *CAPG*, *TNC*, *VCL,* and *PXN*) and F-actin disassembly (*ANXA1*, *ANXA2*, *ANXA7*, *CFL1,* and *PFN1*). The increased *MYL6*, *MYL9*, *MYL12A, MMP14*, as well as the loss of *MYH10* and *MYH14* (Appendix A) could cause decreased focal adhesion, subsequent EMT and actin reorganization leading to upregulated *CDH2* and the loss of epithelial prostate cell markers (*OCLN*, *AR*, *KLK3*, *KLK2*, *CDH1*, *CTNNA1,* and *PTEN*). Moreover, elevated *CAV2* and *CAV1* have been shown to enhance RTK internalization and recycling.

We also assessed the Eph-ephrin and Src pathways in patient-derived CTCs using single-cell RNA-seq (Appendix A). In total, 136 circulating tumor cells were collected from nine patients under active surveillance, of whom six had developed biochemical recurrence (Appendix A). While high heterogeneity of expression patterns in these two signaling pathways existed in the CTCs, four subpopulations (clusters A–D, Appendix A) could be identified using t-SNE analysis. The cells in Clusters A and B had higher expression of genes in Eph-ephrin and Src pathways (Appendix A). The percentage of CTCs in the Clusters A and B in non-biochemically recurrent patients was significantly lower than that in biochemically recurrent patients (Chi-square, *p* = 0.013, Appendix A), which indicates that there were higher expressions of partial gene sets of Eph-ephrin and Src pathways present in the CTCs of patients with more advanced pathological status.

Since these DE genes are involved in actin reorganization, cell migration, and invasion, we further examined whether the mRNA levels of these DE genes would have any significance in predicting clinical outcomes of prostate cancer patients. Based on the RNA-seq of 499 tumors from the prostate cancer patient cohort in The Cancer Genome Atlas (TCGA), the alteration of 21 genes, upregulation of 13 genes (e.g., *ITGA2B*, *FYN*, and *PTK3*), and downregulation of 8 genes (e.g., *PTEN*, *CD44*, and *ARAF*) in the Src signaling pathway were associated with tumors diagnosed with high Gleason scores 9–10 (Appendix A). Moreover, high mRNA levels of six DE genes (≥2 standard deviations), including *EPHB2* and *SRC*, appear to predict poor prognosis for disease-free survival in the TCGA patient cohort using Kaplan–Myer estimation analysis (Figure 3D and Appendix A), which was conducted in The cBio Cancer Genomics Portal [54]. The group of patients with high mRNA levels of *EPHB2/SRC* (red line in Figure 3D) exhibited more than two standard deviations of mRNA levels higher than the group of patients with low mRNA levels (blue line in Figure 3D). These taken together showed that EphB2 and Src pathways predict high-grade prostate tumors and poor prognosis in the TCGA prostate cancer patient cohort.

### 2.4. Disruption of the EphB2 and Src Pathways Led to Decreased EGFR Dynamics, Cell Migration, and Invasion in Advanced Prostate Cancer Cells

Predicting poor clinical outcomes of patients, high transcriptional expression in the Eph-ephrin and Src pathways in advanced prostate cancer intrigued us to explore whether the protein expression of the two master genes in these pathways follows the same trend. The results of Western blotting demonstrated that Src protein levels in both metastatic prostate cancer cell lines, PC3 and DU145, were higher than that of the non-metastatic LNCaP and LNCaP-Abl cell lines (Figure 4A). Similarly, Immunostaining of the Src and EphB2 indicated that they are expressed on the cell membrane and in the cytoplasm (Figure 4B,C). To test our hypothesis that upregulated EphB2 and Src promote metastasis, we assessed cell proliferation, migration, invasion, and EGFR dynamics on LNCaP, DU145, and PC3 treated with or without dasatinib, a tyrosine kinase inhibitor. Our results showed that dasatinib dramatically inhibits the proliferation, migration, and invasion of PC3 cells which overexpress EphB2 and Src (Figure 4D and Appendix A), which agrees with other studies [55,56,57]. Furthermore, dasatinib also reduced the EGFR diffusivity in PC3 cells but had minimal or no effect on the LNCaP cells exhibiting low levels of EphB2 and Src (Figure 4D). DU145 treated with dasatinib showed some slight and moderate reductions in cell migration and invasion (Figure 4D and Appendix A, migration panel, significant difference showed between 9–27 h). However, there is no change in EGFR diffusivity between treated and untreated DU145. Dasatinib blocks a number of tyrosine kinases such as BCR-ABL, the Src kinase family, and other Eph receptors [58]. To achieve a more specific knockdown, we treated cells with small interfering RNA (siRNA) cocktails which specifically targeted *EPHB2* and *SRC* mRNAs.

### 2.5. Downregulated EPHB2 and SRC Attenuated Cell Motility, Invasion, and EGFR Diffusivity in Advanced Prostate Cancer Cells

In prohibition experiments with siRNAs of *SRC* and *EPHB2*, the cell motility and invasive capabilities of metastatic prostate cancer cells were attenuated (−15% in cell motility with siEPHB2 treatment, −49% in cell invasion with siSRC treatment). These were consistent with the reduced EGFR diffusivity (*D* decreased ~30%) and inhibited cortical actin reorganization (*L* decreased 15%). The siRNA cocktails targeting *SRC* and *EPHB2* mRNAs effectively knocked down mRNA levels in both DU145 and PC3 cells compared to the siNon-target control RNA (siNT) (Figure 5A). The SIM images showed that the knockdown of either *SRC* or *EPHB2* in PC3 cells reduces stress fibers but enhances cell–cell adhesion (Figure 5B) and the density of cortical actin (Figure 5B,C). However, in DU145, the knockdown of *EPHB2* reduced stress fibers but had a limited impact on cortical actin (Figure 5B). The quantification of cortical actin was accessed on the basis of fluorescence intensities of xz and yz orthogonal projections along the apical plasma membrane (Figure 5C and Appendix A). In terms of EGFR dynamics, in PC3 cells, the downregulation of *SRC* and *EPHB2* significantly reduced the EGFR diffusivity (Figure 5D). Meanwhile, the PC3 cells exhibited ~70% lower motility (at the 20th hour, Appendix A), but little change in invasion after siEPHB2 treatment (Figure 5E,F). The siSRC treatment strongly inhibited PC3’s invasion (*p* < 0.05, Figure 5F) and slightly attenuated its migration. In DU145, the knockdown of *SRC* slightly increased the cortical actin (Figure 5C), leading to the reduction of EGFR diffusivity (*p* < 0.05, Figure 5D). However, the knockdown of *SRC* did not affect either cell migration or invasion of DU145 (Figure 5E,F). Instead, the downregulation of *SRC* significantly inhibited cell invasion in PC3 (*p* < 0.05, Figure 5F). The time-lapse analysis of migration and invasion is shown in Appendix A.

## 3. Discussion

Despite accumulating evidence of the spatial dysregulation of RTKs leading to tumorigenesis and therapeutic resistance [6], the mechanism underlying this phenomenon is still far from being clear. Here we tracked EGFRs and turned EGFR dynamics into a new type of biophysical biomarker to interrogate the relationships among advanced cancer malignancy, cortical actin organization, and EGFR dynamics. We discovered that more advanced malignancy is correlated with increased EGFR diffusivity, enlarged compartments on the plasma membrane, and more active EGFR internalization, which indicates that EGFR spatial regulation does play an important role in EGFR activation and subsequent metastasis. In addition, we demonstrated that EGFR dynamics in advanced prostate cancer cells is tightly modulated by cortical actin organization, and the reorganization of actin networks is coordinating with cell migration and invasion. Most importantly, the Src and EphB2 pathways were identified as the key factors that coordinate EGFR dynamics, actin organization, and cell motility (a schematic illustration, Figure 5G). To the best of our knowledge, EGFR dynamics have never been used to assess advanced prostate cancer cells.

The inward movement analysis clearly demonstrated that more detailed information about the receptors, such as the endocytosis process and intracellular active transport, can be acquired using more sophisticated 3D-SPT techniques [59,60,61,62,63] or two-color SPT techniques [18,64]. While we have previously made advancements with 3D-SPT techniques [13,65] and reported a wealth of information about EGFR trafficking in skin cancer A431 cells [12], 3D-SPT is currently limited by its low throughput (tracking one receptor complex at a time). As a result of the current throughput limitation with 3D-SPT, we have focused our efforts on EGFR dynamics assays based on 2D-SPT [27].

While the EGFR expression level of DU145 is higher than PC3, PC3’s prominent presence of EGFR phosphorylation at tyrosine 1068 and 1173 (Appendix A) implicated mechanisms beyond quantitative causes. Our findings implied that higher EGFR mobility might collaborate with high EGFR expression levels to further increase the EGFR signaling which promotes cell migration and invasion [66,67]. This was consistent with the upregulation of the majority of genes in EGF-EGFR, and its downstream pathways (PI3K-Akt, Rho GTPase, RAP1, and regulation of actin cytoskeleton) in PC3 revealed by our single-cell RNA-seq (scRNA-seq) data on prostate cancer and its prohibition by tyrphostin (AG 1478), which resulted in suppression of migration and invasion (Horning et al., data not shown). Additionally, we have demonstrated previously that PC3 also exhibits increased EMT profiling and metastatic capacity based on single-cell molecular profiling and biophysical analysis [68,69]. Under the treatments of dasatinib, siSRC, and siEPHB2, the PC3 and DU145 cells became less invasive with a significant decrease in *D* and *L*, as well as a denser cortical actin meshwork. This result indicates that the inhibition of *SRC* and *EPHB2* expression reverses the dissociation of cortical actin. EGFR has multiple interactions and our study has added another layer of regulation [45].

Eph-ephrin and Src pathways are downstream to several growth factor receptors including EGFR [3]. We found that *EPH* and *SRC* over-expressions are common in highly aggressive prostate cancer cells, high-grade tumors, and circulating tumor cells from patients with biochemical recurrence, consistent with previous studies [57,70]. Interference of EphB2 and Src signaling using dasatinib and specific siRNAs inhibits the disruption of cortical actin meshwork and attenuates invasion and migration in advanced prostate cancer cells. We observed less obvious or partial effects of the same prohibitory small molecule and siRNAs in DU145 cells, and that might be due to a moderate gene expression pattern of Src and EphB2 pathways in this advanced cancer cell line. The upregulated Rho GTPases and associated effectors in PC3, such as RAC1, are potential targets of EphB2 and Src (Figure 5G and Appendix A) [71,72]. It is likely that they subsequently orchestrate the actin-binding proteins: regulators of assembly and disassembly (CAPZ, CAPN2, CFL1, and PFN1), nucleators (ARPC1A, ARPC2), crosslinkers (ACTN1, FLNA), and actin-membrane linkers (MYH10, MYH14) to form cortical actin network and the stress fibers in advanced prostate cancer cells [73,74]. The actin cortex organization, density, and thickness have been shown to be the critical factors in membrane tension, contractility, and cell mobility [75]. How cortical actin mediates EGFR dynamics is still a poorly understood process and needs further investigation. However, our results indicate a possible feedback regulation of EphB2 and Src pathways on EGFR signaling via the EGFR dynamics on cell membrane and endocytosis. Among the upregulated genes in these two pathways, the transcriptional or proteomic profiles of several (e.g., the 16-gene set, Figure 3C) present in circulating tumor cells may have great potential to stratify the indolent to more aggressive phenotypes of prostate cancer, thereby serving as liquid biopsy biomarkers.

The EGFR dynamics obtained by SPT points out the difference between non-invasive, invasive, and highly-invasive prostate cell lines. Moreover, we have confirmed the correlation between EGFR dynamics and cortical actin structure by characterizing the cytoskeleton organization with SIM imaging. Our results about actin imaging agree well with San Paulo’s research, where they used AFM to measure cell stiffness and visualized actin structure with a confocal microscope on three breast cancer cell lines with different levels of invasiveness [76]. In terms of the mechanical properties of cancer cells, researchers have repeatedly observed that malignant cells exhibit a more compliant phenotype than their healthy counterparts using prostate cells [68], hepatoma cells [77], and HeLa cells [78,79]. As cells become malignant, mechanical changes have been attributed to an increasingly disorganized cytoskeleton and less pronounced cortical actin. We believe that the denser cortical actin is the key component to constrain the diffusion of EGFRs, and the correlation between membrane receptor dynamics and cortical actin structure enables us to use EGFR dynamics as an indicator to evaluate the organization of cortical actin and to quantify the metastatic potentials of cancer cells.

## 4. Materials and Methods

### 4.1. Cells, Antibodies, and Chemicals

Prostate cancer (PC) cells, LNCaP, DU145, PC3, BPH1 were purchased from ATCC. LNCaP-Abl, an androgen-independent cell line was a gift from Dr. Qianben Wang (The Ohio State University). The cell identity of all cell lines used here was confirmed genetically by The Characterized Cell Line Core at MD Anderson Core. Cells were tested every month for mycoplasma contamination by PCR and always came back found to be negative. Prostate cell lines (PC3, DU145, and LNCaP) were grown in RPMI1640 (22400-089, Thermo Fisher Scientific, Waltham, MA, USA) supplemented with 10% fetal bovine serum (16140-071, Thermo Fisher Scientific) and 1% P/S (15140122, Thermo Fisher Scientific). LNCaP-Abl was cultured in phenol-free RPMI 1640 and 10% charcoal-treated FBS and 1% P/S. Antibodies against SRC1 (2109, Cell Signaling Biotechnology, Danvers, MA, USA), EPHB2 (AF467-SP, R&D Systems), EGFR (MS-311, Thermo Fisher Scientific), and secondary antibodies with FITC and Alexa 530 (712-585-153, Jackson ImmunoResearch Laboratories, West Grove, PA, USA) were purchased commercially. For EGFR tracking in cell monolayers, cells were seeded onto optical imaging 8-well chambered coverglasses (155411, Thermo Scientific, Waltham, MA, USA) with a cell density of 1 × 10^5^ cells per well and allowed to adhere overnight. Latrunculin B (LatB, ab144291, Abcam, Cambridge, MA, USA) was 200 nM in serum-free RPMI medium to depolymerize actin filaments [48] for 10 min.

### 4.2. Fluorescent Probe Labeling to EGFR

Anti-EGFR antibody-conjugated fluorescent nanoparticles were used to label EGFR for tracking. The FN-IgG probe was prepared as described previously [12,13]. Biotinylated monoclonal anti-EGFR antibodies (MS-311-B, Thermo Fisher Scientific, Waltham, MA, USA) were mixed at 1:1 ratio with 40 nm NeutrAvidin-labeled red fluorescent nanoparticles (F8770, Thermo Fisher Scientific, Waltham, MA, USA) in 1.5% BSA/PBS solution (BSA, S7806, Sigma-Aldrich, St. Louis, MO, USA), respectively. The antibody-conjugated fluorescence nanoparticles (~30 nM, the stock solution) can be stored at 4 °C for up to 1 week. The number of antibodies per nanoparticle should follow a Poisson distribution. The brightness of single fluorescent nanoparticles was characterized using both fluorescence correlation spectroscopy and the TSUNAMI microscope. The photon count rates of the 40 nm red fluorescent beads were in the range of 200–500 kHz. For EGFR tracking, cells were seeded onto optical imaging 8-well chambered coverglasses (155409, Thermo Fisher Scientific, Waltham, MA, USA) with a cell density of 1 × 10^5^ cells per well and allowed to grow to ~70% cell confluency. Before tracking experiments, cells were stained with a mixture of Hoechst 33258 (H3569, Thermo Fisher Scientific, 1:1000 dilution in DMEM) and CellMask™ Deep Red (C10046, Thermo Fisher Scientific, 1:1000 dilution in complete medium) for 10 min at 37 °C. After membrane staining, the staining buffer was replaced with the EGFR-labeling solution (antibody-conjugated fluorescent nanoparticles at 100 pM) diluted from the stock solution (30 nM). The reaction was incubated for 30 min at 37 °C. The EGFR-labeling solution was removed, and the samples were washed twice using PBS to remove the unbound fluorescent nanoparticles. Upon completion of membrane staining and EGFR labeling, the chambered coverglass was immediately put on the TSUNAMI microscope for tracking experiments. Two to four trajectories (duration ranged from 1–10 min) were typically obtained from each well. The volumes of all solutions and washing buffers used in staining were 200 µL per well.

### 4.3. D Single-Particle Tracking and Trajectory Analysis

To assess EGFR dynamics, we conducted single-particle tracking (SPT) on the fluorescently labeled EGFRs, and the trajectories were then analyzed using a modified mean-squared displacement fitting algorithm [41,42], generating an averaged EGFR diffusivity (*D*) and a size of the linear confinement (*L*) [12] for each cell line.

Wide-field imaging for SPT is performed using an Olympus IX71 inverted microscope equipped with a 60× 1.2 N.A. water objective (UPLSAPO 60XW, Olympus, Valley, PA, USA). All imaging was conducted at 37 °C using a temperature-controlled stage (Stable Z System, Bioptechs, Butler, PA, USA). Wide-field excitation was provided by a metal halide lamp with a 545/25 nm BP excitation filter. Emission was collected by a Scientific CMOS camera (ORCA-Flash4.0, Hamamatsu, San Jose, CA, USA) through 565 nm dichroic and 605/70 BP. The pixel size is equivalent to 107 nm. Images of FN-IgG tagged EGFRs were acquired at 20 frames per second for a total of 1200 frames. The analysis of the acquired image series was performed as described previously [80,81] to obtain trajectories. The SPT software was a gift from Prof. Keith Lidke at the University of New Mexico. The trajectories were analyzed using MSD analysis to extract *D* and *L*.

All data analysis and image processing were performed within the MATLAB (The Mathworks Inc., Torrance, CA, USA) environment, including the DIPImage image processing library [82]. The images were taken with HCImage (a Hamamatsu’s image acquisition and analysis software), and the single-particle trajectories were reconstructed by using a single-particle tracking software developed by Lidke’s group [81] (see Method S1 for a detailed description). Later, the trajectories were processed into mean-squared displacement curves where the diffusion coefficient (*D*) and the linear dimension of the compartment (*L*) were extracted by curve fitting (Method S2).

### 4.4. In Vitro Cell Proliferation, Migration, and Invasion Assays

Dasatinib (Des-6-[4-(2-hydroxyethyl)-1-piperazinyl]-6-chloro Dasatinib) was purchased from Santa Cruz (sc-500698) and 0.1 µM was used for drug treatment. The siRNA cocktails targeting EPHB2 (ON-TARGETplus SMARTpool, L-003122-00) and SRC (L-003175-00) and control siRNAs (Non-Targeting pool, D-001810-10-05) were obtained from Dharmacon. About 10 K cells were seeded in 96-well culture plates with at least triplicates for proliferation assay. About 60–80 K cells were seeded in ImageLock 96-well plates (4379, Sartorius, Arvada, CO, USA) for migration and invasion and after 48 h the cells reaching 70% confluence were transfected with 5 pmol siRNAs using Lipofectamine 2000 (11668030, Thermo Fisher Scientific) for 5 h. The transfection medium was replaced with fresh normal medium after two gentle washes and adding the drug- or siRNA-media. The confluent cells were scratched with the wound maker (4493, Sartorius). For the invasion assays, a layer of 50 µL of 4 mg/mL Matrigel (354234, Corning, Corning, NY, USA) was added to the scratched cell monolayer. The effects of the small molecule inhibitor and siRNAs on proliferation, migration, and invasion in LNCaP, PC3, and DU145 cells were tracked, measured, and analyzed using the IncuCyte^®^ Zoom system (Essen BioScience, Ann Arbor, MI, USA) according to manufacturer’s protocol as described previously [43].

### 4.5. Cell Fluorescence Imaging Using Structured Illumination Microscopy

Cells were grown on an optical imaging 8-well chambered coverglass (155409, Thermo Scientific, Waltham, MA, USA), fixed with 4% formaldehyde (F8775, Sigma-Aldrich), and permeabilized with 0.1% Triton X-100/PBS (Triton X-100, T8787, Sigma-Aldrich, St. Louis, MO, USA) prior to blocking with 1% BSA in PBS. Then, the samples were incubated with antibodies overnight at 4 °C. To visualize EGFRs, EGFRs were recognized by Anti-EGFR Affibody^®^ Molecule FITC (ab81872, Abcam, Cambridge, MA, USA). The Alexa Fluor^TM^ 633 Phalloidin (A22284, Thermo Fisher Scientific, Waltham, MA, USA) was used to stain F-actin. The fluorescence imaging was taken by Elyra S.1 Structured Illumination Super-Resolution Microscope (SR-SIM) with a 63× 1.2 N.A. water objective. The SR-SIM system is equipped with four excitation laser wavelengths for standard fluorophores (405 nm, 488 nm, 561 nm, and 633 nm) as well as emission filters for DAPI, GFP, RFP, and Alexa 633.

The quantification of the fluorescence intensity of cortical actin was conducted using the software ZEN 2.3 lite. The xz and yz projections of 2 µm width on the xy plan were extracted from the Z-stack of SR-SIM images. The fluorescence intensity of actin along the apical side of the plasma membrane and the brightest region were measured. To make a fair comparison among batches of images, the measured intensities were normalized with the average intensity of the brightest region in the same projection. The average of the normalized fluorescence intensity of actin was taken as an indicator of the density of cortical actin. The procedure of image-based quantification of cortical actin is summarized in Appendix A, and representative fluorescence profiles are presented in Appendix A.

### 4.6. Single-Cell SMART-seq2 (scSMART-seq2) Gene Profiling and Pathway Analysis

To investigate the transcriptomic differences between the LNCaP, LNCaP-Abl, and PC3 cell lines, and circulating tumor cells from patients, the cells were isolated for transcriptomic analysis using SMART-seq2 as previously described [43] (Horning et al., unpublished data). We used the SCDE package from R to adjust for the batch effect and to identify differentially expressed genes between the cell lines [43]. To determine the probability of the observed overlap between the gene expression differences occurring by chance, we used the hypergeometric distributions in R (R: The Hypergeometric Distribution). We correlated the same number of randomly selected reads from the single cells we collected the reads from the bulk cell population. Enriched pathway analysis was performed based on differentially expressed (DE) genes derived from scRNA-seq2 data using Over-representation Analysis algorithm in ConsensusPath Database [83,84]. The raw RNA-seq data circulating tumor cells was deposited in GEO under accession GSE115501.

### 4.7. qRT-PCR

Prostate cancer bulk RNA from four cell lines was harvested with the TRIzol Reagent (15596026, Thermo Fisher), isolated with an RNA MiniPrep Kit (R2052, Zymo Research, Irvine, CA, USA). The cDNA synthesis and gene expression profiling were carried out as described previously [3]. The primer sequences were listed in Appendix A.

### 4.8. Western Blot Analysis

About 100 µg of cell lysates were resolved on Bolt™ 4–12% Bis-Tris Plus Gels (NW04120BOX, Thermo-Fisher Scientific). The separated proteins were transferred on to Odyssey^®^ Nitrocellulose Membranes (LI-COR, Lincoln, NE, USA) using a Mini-PROTEAN^®^ 3 Electrophoresis System (Bio-Rad, Hercules, CA, USA) filled with 800 mL of transfer buffer and run at 20 V for 10 h at 4 °C using a PowerPac Basic (Bio-Rad). The membranes were blocked with 5% milk in TBST for 1 h. The SRC, EPHB2, and GAPDH bands were probed with specific antibodies and then secondary antibodies. The bands were visualized after being incubated in 5% milk in TBST with 1:2000 goat anti-rabbit secondary antibody (sc-2004, Santa Cruz Biotechnology, Santa Cruz, CA, USA) for 1 h. The membranes were put in between two pieces of plastic wrap and loaded into an Amersham Hypercassette (GE Healthcare Life Sciences, Pittsburgh, PA, USA). Membranes were exposed with HyMembrane CL Autoradiography Film (Denville Scientific, Metuchen, NJ, USA) to obtain images that were subject to scanning and then densitometric quantification for protein expression using ImageJ [85].

### 4.9. Patients and Ethical Approval

The single-cell circulating tumor cell protocol has been approved by institutional review board of the University of Texas Health Science Center at San Antonio (#CTRC 130001). Nine patients were enrolled (Appendix A). Written informed consents were obtained from all patients.

### 4.10. Statistical Analysis

For all groups that were statistically compared, the equality of variances was tested by F-test. Comparisons between two groups were calculated using unpaired Student’s *t*-test (two tailed), and *t*-test with equal variances or unequal variances were selected according to the results of F-test. The statistical significances of the unpaired *t*-tests were set as *p* < 0.05, *p* < 0.01, and *p* < 0.001 for two-tailed *t*-test. Results are presented as means ± standard errors. Comparisons between more than two groups were calculated with one-way analysis of variance (ANOVA). Rather than a normal distribution, lognormal distribution was often used to describe the broad distribution of particle-trajectory-derived diffusivity [86,87]. The central limit theorem can be applied in the *t*-tests because of the great numbers of trajectories we collected in our study (at least 300 trajectories), and the *t*-test is valid even when *D* and *L* follow a lognormal distribution. The *t*-test is based on the two groups means $\bar{X_{1}}$ and $\bar{X_{2}}.$ Because of the central limit theorem, the distribution of $\bar{X_{1}}$ and $\bar{X_{2}}$, in repeated sampling, converges to a normal distribution, irrespective of the distribution of *X* in the population [88]. Thus, *t*-test is able to be used to test *D* and *L* among cell lines or cell lines with different treatments.

## 5. Conclusions

We discovered that EGFR dynamics could be used to differentiate the highly invasive cancer cell line (PC3) from the non- (LNCaP) and less (DU145) invasive cancer cell lines. Through the combined interpretation of SPT and SIM imaging, we revealed that cortical actin modulates the dynamics of EGFRs. Finally, we demonstrated a strong correlation between the EGFR dynamics and the metastatic potential that was quantified by measuring cell migration, invasion, and proliferation. When treated with dasatinib or siRNA, EGFR dynamics were closely correlated with the changes to migration and invasion through SRC and EPH signaling. Thus, we believe that the SPT-based EGFR dynamics can serve as a new biophysical assay to probe the metastatic potential of cancer cells and to monitor their response to anti-cancer drug treatments.

## Figures and Tables

**Figure 1 cancers-11-01910-f001:**
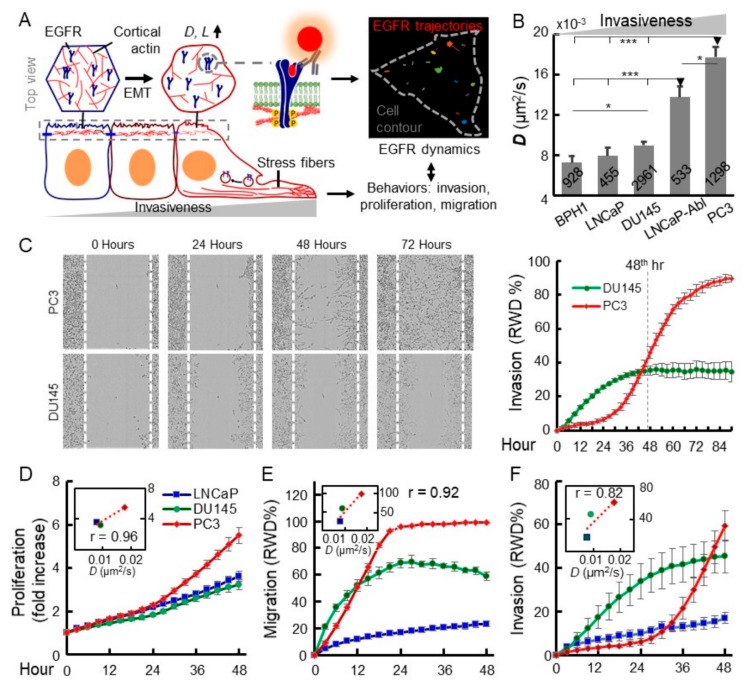
Increased epidermal growth factor receptor (EGFR) dynamics shown by Transmembrane Receptor Dynamics (TReD) are strongly correlated with metastatic malignancy in prostate cancer cell lines. (**A**) Our hypothesis that, during metastasis, the reorganization of actin network facilitates cell invasion, migration, and EGFR diffusivity. The positive correlation between EGFR diffusivity and cancer malignancy would allow us to develop a biophysical phenotyping assay for quantifying malignancy of prostate cancer cells. (**B**) Characterization of EGFR diffusivity among five prostate cell lines. The highly-invasive PC3 cell line exhibits significantly higher EGFR diffusivity than non- or less invasive cells. The number of trajectories collected from each cell line is labeled on each bar. Typically, 5–15 trajectories were collected from every single cell. All statistical analyses were performed using the unpaired *t*-test. The asterisk represents the level of statistical significance for *t*-test: * *p* < 0.05, *** *p* < 0.001. The error bar represents the standard error of the mean. (**C**) Representative images of scratched cell monolayers and migrating cells at the 0th, 24th, 48th, and 72nd hour in the invasion assay. The cell invasion of PC3 and DU145 cells were monitored, measured, and analyzed using the IncuCyte^®^ Zoom system. RWD stands for relative wound density. The RWD of DU145 achieved a plateau at around hour 40, but the RWD of PC3 kept increasing until around the 80th hour. This result indicates PC3 is more invasive than DU145. (**D**–**F**) Proliferation, migration, and invasion of three prostate cancer cells. The insets are the scatter plots of EGFR diffusivity versus proliferation and cell motility at *t* = 48 h. Their corresponding Pearson correlation coefficients (*r*) are labeled in each plot inlet.

**Figure 2 cancers-11-01910-f002:**
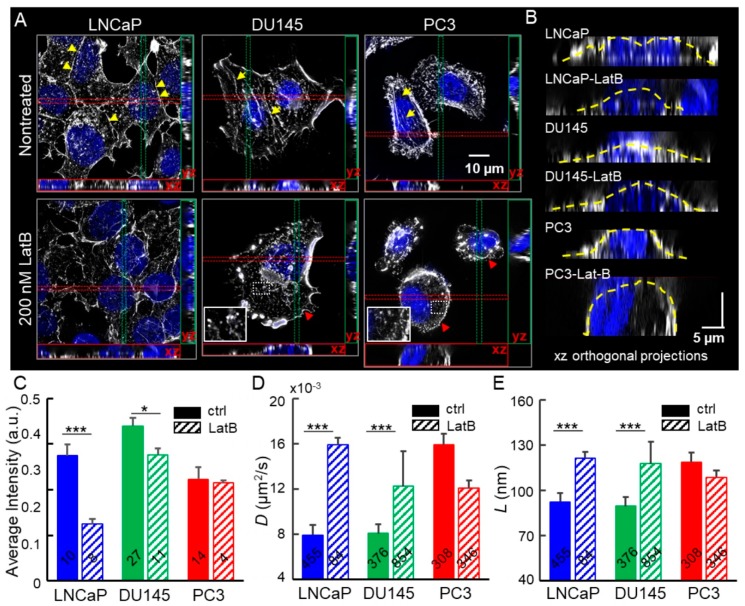
Depolymerization of F-actin induces increased EGFR diffusivity and compartment size. (**A**) Maximum intensity projection on the xy plane, and orthogonal cross-sections (xz and yz) of LNCaP, DU145, and PC3 cells before and after Latrunculin B (LatB) treatment. LatB inhibits the polymerization of F-actin. The yellow arrowheads, yellow arrows, and red arrows pinpoint the peri-junctional actin bands, the stress fibers on the basal side of cells, and the filopodia, respectively. (**B**) The xz projections clearly show the dissociation of cortical actin from the apical surface of the plasma membrane after LatB treatment. (**C**) Quantification of cortical actin based on fluorescence intensities of xz and yz orthogonal projections along the apical plasma membrane (shown as the yellow dashed line in (**B**)). The fluorescence intensities are normalized and presented as an arbitrary unit (a.u.). The number of projections analyzed is labeled on each bar. (**D**) Diffusion coefficients of EGFRs extracted from trajectories. The number of trajectories analyzed is labeled on each bar. (**E**) The linear dimension of confinement extracted from EGFR trajectories. All statistical analyses were performed using the unpaired *t*-test. The asterisk represents the level of statistical significance for *t*-test: * *p* < 0.05, *** *p* < 0.001. The error bar represents the standard error of the mean.

**Figure 3 cancers-11-01910-f003:**
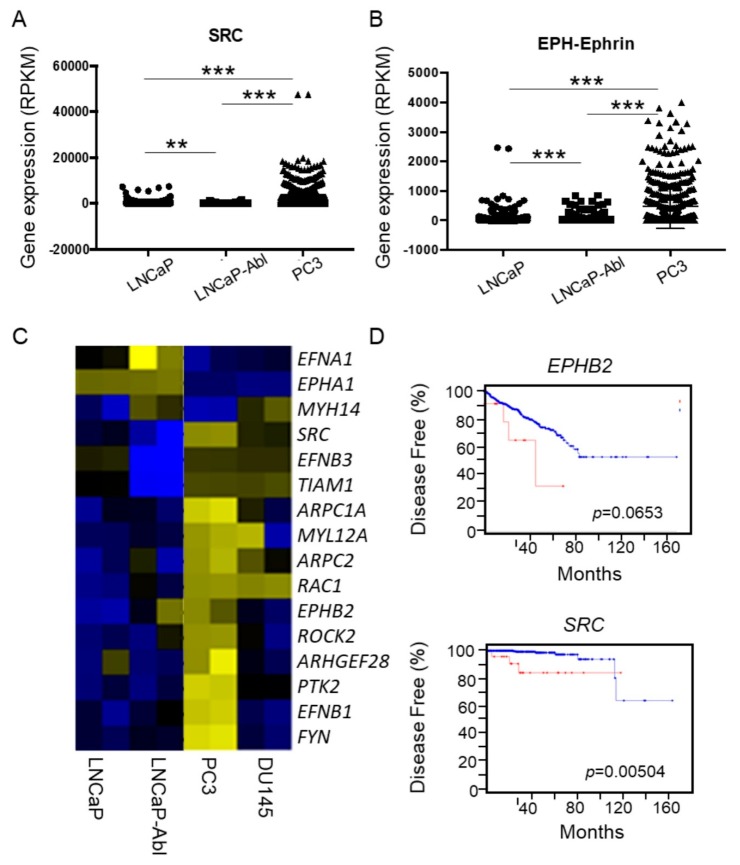
Single-cell RNA-seq showing upregulation of Src and Eph-ephrin signaling in advanced prostate cancer cells. (**A**) Dot plot of differentially expressed (DE) genes (n = 171) in Src signaling. LNCaP, LNCaP-Abl, and PC3 are clusters distinctly separated from the other clusters. (**B**) Dot plot of DE gene expression (n = 20) in Eph-Ephrin signaling. Unpaired *t*-test, ** *p* < 0.01, *** *p* < 0.0001. (**C**) The expression of 16 representative genes from EPH Transcripts of 16 genes in four cell lines were validated using bulk RNA qRT-PCR. (**D**) mRNA levels of *EPHB2* and *SRC* predict poor prognosis in The Cancer Genome Atlas (TCGA) prostate cancer patient cohort using Kaplan–Myer estimation analysis. The red line represents the patients with a high expression of the genes with more than 2 SD as compared to the patients presented in the blue line.

**Figure 4 cancers-11-01910-f004:**
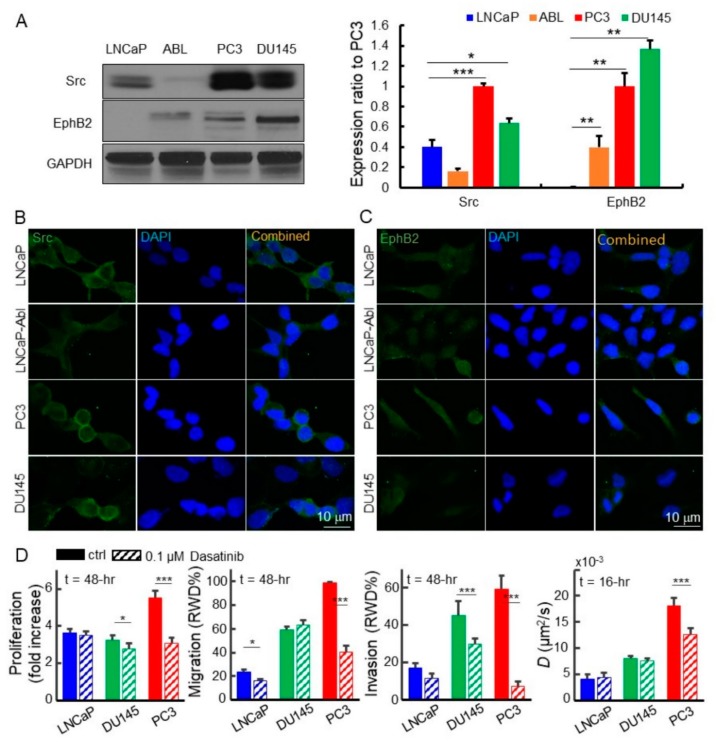
Dasatinib inhibits proliferation, migration, invasion, and EGFR diffusivity in advanced prostate cancer cells. (**A**) Src is highly upregulated in PC3 (2.5x) and DU145 (1.6x) as compared to LNCaP shown in Western blots. EphB2 is overexpressed in both PC3 and DU145. Both proteins are almost not expressed in LNCaP-Abl. (**B**) Src is present in these cell lines. There is an intense level of Src on the PC3 cell membrane. (**C**) Immunostaining of EphB2 protein is present in plasma and membrane. (**D**) Image-based IncuCyte assays allow us to conduct the time-lapse analysis of cell proliferation, migration, and invasion of the cells treated with or without dasatinib. The dasatinib significantly inhibits the proliferation, migration, and invasion of DU145 and PC3 cells but reduces the EGFR diffusivity in only PC3 cells. The mean value of each bar was measured at the end time of each assay or at the 48th hour. All statistical analyses were performed using the unpaired *t*-test. The asterisk represents the level of statistical significance for *t*-test: *** *p* < 0.001, ** *p* < 0.01, * *p* < 0.05. The error bar represents the standard error of the mean. The RWD stands for relative wound density.

**Figure 5 cancers-11-01910-f005:**
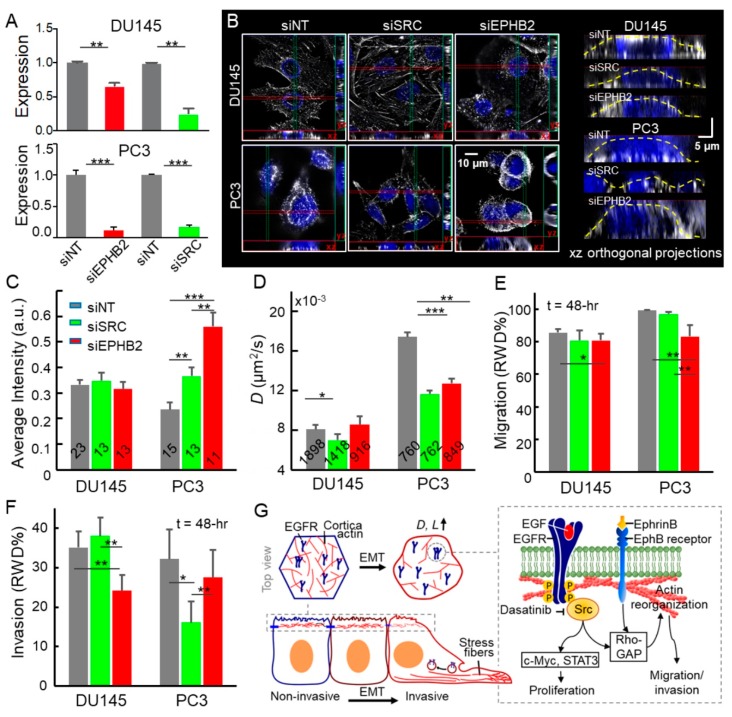
Disruption of EphB2/Src pathways leads to attenuated cell motility, invasion, and EGFR diffusion in advanced prostate cancer cells. (**A**) Effective gene knockdowns in siRNA-treated DU145 and PC3. (**B**) Structured Illumination Microscopy (SIM) images of siRNA treated cells. Maximum intensity projection on the xy plane and orthogonal cross-sections (xz and yz) of DU145 and PC3 siRNA treated cells. (**C**) Quantification of cortical actin based on fluorescence intensities of xz and yz orthogonal projections along the apical plasma membrane. The number of projections analyzed is labeled on each bar. (**D**) EGFR diffusivities of the siRNAs treated cells. The error bar represents the standard error of the mean. (**E**,**F**) The image-based assays allow us to conduct the time-lapse analysis of cell migration and invasion on the siRNA-treated cells. The error bar represents the standard deviation. All statistical analyses were performed using the unpaired *t*-test. The asterisk represents the level of statistical significance for *t*-test: *** *p* < 0.001, ** *p* < 0.01, * *p* < 0.05. (**G**) Schematic shows the effects of EMT-induced actin reorganization on EGFR dynamics and the Src/EphB2 induced signaling from the plasma membrane that controls cell behavior.

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
