# Peer review of "Spatial EGFR Dynamics and Metastatic Phenotypes Modulated by Upregulated EphB2 and Src Pathways in Advanced Prostate Cancer"

_cancers, 2019, doi:10.3390/cancers11121910_

Round 1

Reviewer 1 Report

This is a very well scientifically conceived and perfectly executed experimental design that provides strong rationale and logistical progression in conducting this study. This study has provided valuable information with regards to moving the field in exploring the molecular underpinnings of prostate cancer metastasis and EGFR signaling dynamics. It would be highly appreciated by readers if authors could briefly mention the potential of 16 signature of Figure 3C to serve a future polygenic risk score to stratify the indolent (watchful waiting) to more aggressive phenotypes of prostate cancer, could these 16 gene signature along with other proteomic profiles help delineate prostate cancer progression in vivo mileau as strongly exhibited in patient derived cell lines in vitro model systems.

Author Response

We thank the reviewers for their comments and feel that the revised manuscript is much stronger due to their efforts. The point-to-point responses listed below, and a response letter is attached. 

Comment 1: It would be highly appreciated by readers if authors could briefly mention the potential of 16 signature of Figure 3C to serve a future polygenic risk score to stratify the indolent (watchful waiting) to more aggressive phenotypes of prostate cancer.

Response 1: We thank the reviewer for the suggestion. A sentence was added in the Discussion Section to point out the potential of using these 16 genes as biomarkers to differentiate indolent subtype from more aggressive prostate cancer. The added discussion in the main text (line 406-409) is reproduced below:

…Among the upregulated genes in these two pathways, the transcriptional or proteomic profiles of some genes (e.g. the 16-gene set, Figure 3C) in circulating tumor cells may have great potential to stratify the indolent to more aggressive phenotypes of prostate cancer and serve as liquid biopsy biomarkers.

Comment 2: Could these 16-gene signatures along with other proteomic profiles help delineate prostate cancer progression in vivo milieu as strongly exhibited in patient-derived cell lines in vitro model systems.

Response 2: This is a very good suggestion since the 16-gene signatures may have great potential for prognosis prediction and stratifying indolent or castration resistant patients. We envision that the integration of transcriptional and proteomic profiles will further improve the accuracy of prognosis prediction, which is exactly the project we are working on. The response to this comment can be found in discussion in the main text (line 406-409) is reproduced below:

…Among the upregulated genes in these two pathways, the transcriptional or proteomic profiles of some genes (e.g. the 16-gene set, Figure 3C) in circulating tumor cells may have great potential to stratify the indolent to more aggressive phenotypes of prostate cancer and serve as liquid biopsy biomarkers.

Reviewer 2 Report

In this manuscript, the possible interaction between EGFR, SRC and EphB2 is investigated. I found the used methodology for imaging interesting and the microscopic data are so attractive. Also, the TCGA data and the expression data of this research are in good agreement. Functionally, there results are solid while mechanistically the results are not as it claims in the abstract (line 47-48).  

In my review, this manuscript could be considered for publication due to the used interesting methodology of imaging and he reported functional data. However, I'd like the following revision:

Author may make the manuscript easier to follow and remove claims with no solid evidence for them. 

Figure 1A needs more elaboration. 

The Heatmaps on Fig 3A should be moved to the supplemental data and it should be replaced with a simple correlation expression chart.

I found many formatting issues such as inconstancy between the used labels on the Figs (A,B,C, ...) and the legends (a, b, c, ...)

In all, the manuscript should be more simple and the data which do not support the core results should be moved to supplemental section.   

Author Response

We thank your comments and suggestions and feel that the revised manuscript is much stronger due to their efforts. Please see the attached response letter in which we addressed your comments point-by-point. Thanks. 
